# SMA20/PMIS2 Is a Rapidly Evolving Sperm Membrane Alloantigen with Possible Species-Divergent Function in Fertilization

**DOI:** 10.3390/ijms25073652

**Published:** 2024-03-25

**Authors:** Nathaly Cormier, Asha E. Worsham, Kinsey A. Rich, Daniel M. Hardy

**Affiliations:** 1Department of Biological Sciences, University of Wisconsin-Whitewater, Whitewater, WI 53190, USA; 2Department of Cell Biology & Biochemistry, Texas Tech University Health Sciences Center, Lubbock, TX 79430, USA; asha.worsham@ttuhsc.edu (A.E.W.); kinsey.rich@ttuhsc.edu (K.A.R.)

**Keywords:** fertilization, spermatozoa, cell membrane, molecular evolution, alloimmunity, genome annotation, de novo gene

## Abstract

Immunodominant alloantigens in pig sperm membranes include 15 known gene products and a previously undiscovered *M*r 20,000 sperm membrane-specific protein (SMA20). Here we characterize SMA20 and identify it as the unannotated pig ortholog of PMIS2. A composite SMA20 cDNA encoded a 126 amino acid polypeptide comprising two predicted transmembrane segments and an N-terminal alanine- and proline (AP)-rich region with no apparent signal peptide. The Northern blots showed that the composite SMA20 cDNA was derived from a 1.1 kb testis-specific transcript. A BLASTp search retrieved no SMA20 match from the pig genome, but it did retrieve a 99% match to the *Pmis2* gene product in warthog. Sequence identity to predicted PMIS2 orthologs from other placental mammals ranged from no more than 80% overall in Cetartiodactyla to less than 60% in Primates, with the AP-rich region showing the highest divergence, including, in the extreme, its absence in most rodents, including the mouse. SMA20 immunoreactivity localized to the acrosome/apical head of methanol-fixed boar spermatozoa but not live, motile cells. Ultrastructurally, the SMA20 AP-rich domain immunolocalized to the inner leaflet of the plasma membrane, the outer acrosomal membrane, and the acrosomal contents of ejaculated spermatozoa. Gene name search failed to retrieve annotated *Pmis2* from most mammalian genomes. Nevertheless, individual pairwise interrogation of loci spanning *Atp4a*–*Haus5* identified *Pmis2* in all placental mammals, but not in marsupials or monotremes. We conclude that the gene encoding sperm-specific SMA20/PMIS2 arose de novo in Eutheria after divergence from Metatheria, whereupon rapid molecular evolution likely drove the acquisition of a species-divergent function unique to fertilization in placental mammals.

## 1. Introduction

Gene products expressed only in the gametes mediate the unique cellular events of animal fertilization [1,2,3,4,5,6]. In mammals, many sperm-specific proteins are known or suspected to function in fertilization events, ranging from sperm transport in the female tract to ultimate fusion of sperm and egg plasma membranes [7,8,9,10,11,12,13,14,15,16,17,18,19]. Nevertheless, it is unclear whether past studies, conducted in diverse mammalian species, have collectively identified all, most, or only some of the key proteins that mediate fertilization. Sperm-specific proteins are, by definition, “non-self” in females, so they typically evoke a robust alloimmune response in conspecific females [11,20,21,22], whereas proteins common to somatic cells do not. Accordingly, to search for yet-unknown fertilization proteins, we systematically identified immunodominant alloantigens in pig sperm membranes and lipid rafts by targeted proteomics, and thereby defined the repertoire of major sperm-specific membrane proteins in a single species [21,22]. The identified alloantigens included known sperm-specific membrane proteins, known somatic cell proteins not previously shown to be expressed as alloantigenic, sperm-specific isoforms, and one protein with peptide sequences that yielded no matches to sequences in the NCBI non-redundant protein database [22]. This potentially novel protein migrated with Mr 20,000 in SDS-PAGE and was designated “Sperm Membrane Alloantigen 20” (SMA20). Here, we report the cDNA sequence, tissue-specific expression, localization, species diversity, and genomic ontogeny of SMA20. Our findings provide insight into genetic processes, including genome rearrangements, de novo gene origination, and rapid molecular evolution, that give rise to gamete-specific proteins with unique functions in fertilization.

## 2. Results

### 2.1. SMA20 cDNA Cloning and Protein Sequence Analysis

We first cloned cDNA fragments spanning the full SMA20 coding sequence by a combination of 3′- and 5′-RACE (Figure 1 and Appendix A).

A 436-bp 3′-RACE product amplified using nested, degenerate sense primers encoding the amino acid sequence WEEAYiN (i = I or L; Appendix A) comprised a poly-A tract and an upstream ORF that encoded de novo peptide sequence not specified by the sense primer, verifying that the cloned cDNA was an authentic 3’-end amplicon of the SMA20 mRNA. Two subsequent rounds of 5′-RACE using six different specific reverse primers spanning nucleotides 15–338 of the SMA20 3′-end cDNA fragment (Appendix A) amplified 5′-end SMA20 cDNAs spanning the 5′-end of the ORF and 5′-UTR. The assembly of the 3′- and 5′-RACE product sequences yielded a contig spanning the full SMA20 ORF (Figure 1A) with overlap at nucleotides 359–432. To confirm the composite sequence, we amplified a full-length SMA20 cDNA by RT-PCR using a mix of *Taq* and *Pfu* polymerases, which produces high-fidelity amplicons, owing to 3′ exonuclease activity of *Pfu* polymerase [25]. A 3′-end consensus sequence assembled from 17 individual reads, together with a 5′-end consensus sequence from 12 reads, yielded a definitive, full-length sequence identical to the 786 bp composite sequence from the RACE fragment assembly.

The 5′ end of the full-length SMA20 cDNA encompasses a 75 nt 5′-UTR (nts 1–75) with two upstream, in-frame stop codons (at nts 52–54 and 61–63, respectively), followed by an initiator methionine codon (nts 76–78) in a good Kozak consensus [23]. The 378 nt ORF (nts 76–453) encodes a 126-residue polypeptide terminated with a TGA stop codon (nts 454456). The 3′ end of the cDNA encompasses a 330 nt 3′UTR (nts 457–786) with a non-canonical polyadenylation signal (ATTAAA) at nts 760–765 and a poly(A) tail.

The sequence analysis of the 126 amino acid deduced that polypeptide predicts a molecular mass of 13,294.47 Da and an isoelectric point (pI) of 4.7, and it identifies two putative transmembrane segments (designated TMS1 and TMS2) at positions 61–81 and 107–126, respectively (Figure 1B,C), consistent with an integral membrane protein. The N-terminus of SMA20 lacks a cleavable signal peptide per the criteria of von Heijne [24] and, instead encodes an alanine- and proline-rich region with two putative Src homology 3 (SH3) binding motifs (PGAP) at positions 24–27 and 36–39, as well as multiple predicted sites for post-translational modifications (glycation, O-glycosylation, and phosphorylation). The two predicted transmembrane segments flank an intervening, hydrophilic “loop” also with putative, post-translational modification sites (Figure 1C).

### 2.2. Tissue Distribution of the SMA20 Transcript

On Northern blots of pig total RNAs, a 5′-end probe (nucleotides 1–395; Figure 1A) detected a most abundant SMA20 transcript migrating at 1.1 kb in testis (Figure 2), but there was no apparent expression in twelve other pig tissues (Figure 2A) or in the testis of sexually immature boars (Figure 2B). Oligonucleotide probes to 5′- and 3′-end sequences both hybridized with similar intensity to the 1.1 kb transcript (Figure 2C), confirming that the 1.1 kb band corresponds to a unique full-length SMA20 transcript in mature testis.

### 2.3. Characterization of SMA20 in Sperm Membrane Fractions

Kyte–Doolittle, Emini, and Jameson–Wolf algorithms predicted both the loop and P-rich regions of SMA20 to be highly hydrophilic, surface exposed, and antigenic, respectively (Figure 1B). Monospecific peptide antibodies (serum and affinity-purified) to the loop region yielded strong and specific immunoreactivity at M*r* 20,000 on Western blots of sperm membrane proteins (Figure 3A), consistent with SMA20 detection by alloantisera. Anti-P-rich-region antibodies also yielded major immunoreactivity at Mr 20,000, as well as minor immunoreactivity at M*r* 37,000 (Figure 3A). Preabsorption of anti-loop and anti-P-rich sera and affinity-purified antibodies with their cognate peptides strongly diminished the immunoreactivity at both positions (Figure 3B). Consistent with the properties of integral membrane proteins, non-ionic detergent (Triton X–100) quantitatively extracted SMA20 protein from spermatozoa (Figure 3C); the re-extraction of Triton-extracted spermatozoa with SDS yielded no additional SMA20 immunoreactivity with either antibody. Furthermore, both SMA20 antibodies detected the protein in the particulate fraction containing mixed vesiculated membranes released from spermatozoa upon the induction of AR by ionophore A23187, but not in the soluble fraction (not shown). Alloantisera raised against “triple-washed” membranes (TWM) and lipid rafts both recognized SMA20 in TWM (Figure 3D), consistent with our original identification of the protein [22].

### 2.4. SMA20 Localization in the Apical Head of Spermatozoa

The anti-P-rich antibody detected strong SMA20 immunoreactivity on the anterior head of MeOH-fixed ejaculated pig spermatozoa, coincident with the acrosome but exclusive of the equatorial segment (Figure 4A). The anti-loop antibody yielded comparatively less immunoreactivity, restricted to the apex of the acrosomal region. Immunopositive cells in all ejaculates tested (*n* = 8) produced the same labeling patterns. Furthermore, for both antibodies, all positive cells also were labeled positive for WGA (specific for the plasma membrane; Figure 4A,B) and PNA (specific for the outer acrosomal membrane; not shown). Conversely, the cells negative for both lectins also yielded no immunoreactivity with either antibody. Preabsorbing the SMA20 antibodies with their cognate peptide immunogens completely ablated immunoreactivity, confirming specificity of the interactions (Figure 4A). Likewise, non-immure rabbit IgGs (Figure 4B) or secondary antibody only also yielded no immunoreactivity. The live labeling of spermatozoa (30 min incubation) yielded little or no immunoreactivity using either antibody (Figure 4B; n = 5); increasing the concentration of the primary antibody and incubation time produced no change in the intensity or incidence of labeling. Regardless of protocol, live-labeling experiments yielded immunoreactivity on only 5–10% of spermatozoa, with patterns similar to those of fixed cells immunolabeled with either anti-loop or anti-P-rich. In the immunofluorescence performed entirely on suspensions of living cells in suspension, both antibodies immunolabeled only non-motile cells, suggesting that SMA20 was accessible to an antibody only on cells with a damaged plasma membrane.

### 2.5. Ultrastructural Localization of SMA20

To gain insight into the function of SMA20, we determined its ultrastructural localization and membrane orientation by immunoelectron microscopy (Figure 5).

The αP-rich antibody detected the SMA20 AP-rich domain predominantly in apposition to the inner leaflet of the peri-acrosomal plasma membrane upon the post-embedding–labeling of ultrathin sections post-fixed with osmium tetroxide (Figure 5A). The αLoop antibody yielded no specific localization signal irrespective of the immunolabeling protocol (pre- or post-embedding, with or without osmium post-fixation). The αP-rich antibody also bound a small number of gold particles to the outer acrosomal membrane and acrosomal matrix of some spermatozoa. To assess the specificity of the sparse localization signal obtained with the αP-rich antibody, we compared it to the signal from parallel sections immunolabeled using antibody preabsorbed with its cognate peptide, and we quantified the distribution of gold particles in 40 randomly chosen microscopic fields. The peptide strongly blocked the plasma membrane-associated labeling (bar graph, Figure 5A).

The immunolabeling of sections without osmium post-fixation modestly increased the sperm head-associated localization signal (Figure 5B). Most gold particles localized to the perimeter of the cell overlying the acrosome, consistent with SMA20 presence in the plasma and outer acrosomal membranes, but with no ability to discriminate between the two membranes owing to the comparatively poorer preservation of membrane ultrastructure. The non-post-fixed labeling protocol also detected immunoreactivity in the acrosomal matrix (Figure 5B). A quantitative comparison to cells immunolabeled with preabsorbed antibodies confirmed the specificity of the apical head-associated labeling pattern (bar graph, Figure 5B). The immunolabeling of parallel sections with antibody to zonadhesin holoprotein yielded the same pattern, in agreement with previous ultrastructural localization of zonadhesin to the perimeter of the acrosome and outer acrosomal membrane of pig spermatozoa [26].

### 2.6. Identification of SMA20 Orthologs and Genomic Loci

BLASTp queries of the NCBI non-redundant protein database with the SMA20 deduced amino acid sequence retrieved a 99% identical match to a warthog gene product annotated as PMIS2 (“Protein Missing In Spermatozoa 2”) but no matches of greater than 80% identity to gene products from any other species, including the pig. A pairwise comparison of the pig SMA20 cDNA sequence with the warthog *Pmis2* gene coding sequence (cds) confirmed the co-linearity of the two sequences (Figure 6).

Word search of the NCBI nucleotide database retrieved the annotated *Pmis2* locus from 86 species of placental mammals, situated between *Atp4a* and *Haus5* and downstream of *Gapdhs* (gene encoding germ cell-specific GAPDH [27]) in all species. In contrast, a comparable word search retrieved annotated *Atp4a*, *Haus5*, and *Gapdhs* from 373, 376, and 294 placental genomes, respectively. Pairwise interrogation of the *Atp4a*–*Haus5* intergenic region with the pig SMA20 cDNA sequence correctly identified the two-exon *Pmis2* gene in five representative species (Figure 6), including the previously un-annotated pig locus, as well as *Pmis2* from armadillo (*Dasypus novemcitus*), a relatively primitive species from superorder Xenarthra that diverged from the other species near the base of the placental phylogeny [28,29]. To determine if *Pmis2* is conserved in Placentalia, we interrogated, via a pairwise comparison to the armadillo *Pmis2* cds, the *Atp4a*–*Haus5* intergenic sequences in the genomes of 20 randomly chosen placental species in which *Pmis2* had previously evaded annotation, and we identified the two-exon *Pmis2* gene in all 20 species. In contrast to the likely conservation of the *Pmis2* locus throughout Placentalia, a word search retrieved no annotated *Pmis2* loci in marsupial or monotreme species, suggesting that the gene is either absent or unrecognizably divergent in these mammalian clades. We therefore conducted a purposeful search for highly divergent SMA20/PMIS2 genes in non-placental mammalian species (Figure 7).

Interrogating genomic loci spanning *Atp4a*–*Haus5* in the armadillo and opossum (*Monodelphis virginiana*) via a pairwise comparison to an armadillo *Atp4a*–*Pmis2*–*Haus5* concatemeric sequence readily identified all exons of *Atp4a* and *Haus5* in the armadillo and opossum, and *Pmis2* in the armadillo (not shown; see Figure 6, armadillo panel) but not the opossum (Figure 7). A comparable interrogation of the genomic locus spanning *Atp4a*-like–*Etv2* in the platypus (*Ornithorhynchus anatinus*) via a pairwise comparison to an armadillo *Atp4a*–*Pmis2*–*Etv2* concatemeric sequence identified *Atp4a*-like and *Etv2* gene exons in the platypus (Figure 7), but no *Pmis2*. Collectively, these analyses readily identified exons comprising the expected flanking genes (*Atp4a* and *Haus5* in the armadillo and opossum, and *Atp4a*-like and *Etv2*-like in the platypus), but only retrieved the two *Pmis2* exons from the armadillo, consistent with presence of the gene in placental mammals but not marsupials or monotremes.

### 2.7. Rapid Evolution of SMA20/PMIS2

To assess species diversity of SMA20/PMIS2, evident initially as discontinuities in exon identification via a dot plot (Figure 6), we compared SMA20/PMIS sequences among a diverse selection of placental mammals (Figure 8).

The N-terminal regions of SMA20/PMIS2 polypeptides varied dramatically among species (Figure 8A), both in length and sequence. Lengths ranged from 70 amino acids in the big brown bat, *Eptesicus fuscus*, to only five in Kuhl’s pipistrelle bat, *Pipistrellus kuhlii*), and sequence differences manifested most strikingly as the general absence of AP-rich sequence in many myomorph rodents (family = Muridae, and genera = *Mus*, *Meriones*, and *Psammomys*; family = Cricetidae, and genera = *Chionomys*, *Microtus*, *Peromyscus*, and *Phodopus*), and the replacement of AP-rich sequence with ST-rich sequence in old-world primates (family = Cercopithecidae, and genus = *Macaca*; family = Hominidae, and genera = *Gorilla*, *Homo*, and *Pongo*; family = Hylobatidae, and genera = *Hylobates* and *Symphalangus*). In contrast, the sequences immediately upstream of and including the transmembrane segments aligned more consistently across species (Figure 8B), but with generally greater frequency of unconservative amino acid substitutions in the blocks of ~20 amino acids preceding TMS1 and TMS2, corresponding to the junction between the AP-rich region of SMA20 and TMS1, and the C-terminal end of the loop region.

### 2.8. Genomic Ontogeny of SMA20/PMIS2

To determine if the absence of *Pmis2* in the *Atp4a*–*Haus5* intergenic segments of platypus and opossum genomes reflected rearrangements that might have disrupted the shared synteny of an ancestral *Pmis2* locus early in the divergence of Mammalia, we compared the gene order and orientation of expanded genomic regions encompassing the seven genes upstream and downstream of *Pmis2* in placental mammals. The analysis identified inversions that occurred since divergence ~220 Myr ago [30] of subclass Prototheria (represented by the platypus in the order Monotremata) and subclass Theria (represented by the opossum in the infraclass Marsupialia, order Didelphimorphia; and by the mouse in infraclass Placentalia, order Rodentia), but no displaced *Pmis2* gene in either the opossum or platypus. Specifically, among the four genes on either side of mouse *Pmis2*, inversion in the therian lineage (Metatheria + Eutheria, represented by opossum and mouse, respectively) of *Tmem147* produced a change in orientation but not gene order, whereas inversion of a three-gene cassette (*Haus5*–*Rbm42*–*Etv2L*) changed the gene order, producing a tail-to-tail orientation of *Atp4a* and *Haus5* (Figure 9). The newly formed *Atp4a*–*Haus5* intergenic segment then appears to have served as raw material for the de novo genesis of *Pmis2* in the placental lineage (Eutheria) after it diverged ~150 Myr ago from Metatheria [28,30].

## 3. Discussion

Our results identify SMA20 as the pig ortholog of PMIS2, a mouse sperm protein previously shown to be missing from mice made deficient in the sperm-specific chaperone calmegin [31]. *Clgn* loss-of-function disrupts the trafficking of multiple sperm-specific membrane proteins, including not only PMIS2 but also ADAM3 (“A Disintegrin And Metalloprotease” 3) [31] and causes near sterility [32] by impairing sperm transport into the oviduct [31]. Likewise, ablation of *Pmis2* or of six other genes (*Calr3*, *Pdilt*, *Tpst2*, *Ace*, *Adam1a*, and *Adam2*) produces the same combined phenotype consisting of near or complete sterility, impaired sperm transport, and loss of ADAM3 in the membranes of mature spermatozoa [31]. Curiously, spermatozoa lacking ADAM3, regardless of the upstream cause, also cannot fertilize cumulus-free mouse eggs, owing to a defect in adhesion to the zona pellucida, but they can readily fertilize eggs enveloped by a normal cumulus cell mass [31,32]. The former scenario is common in IVF studies, whereas the latter is physiologically correct [33], and the ability of the cumulus mass to rescue the adhesion phenotype provides insight into the biochemistry underlying gene disruptions that perturb ADAM3 trafficking. Sperm cells must expose or release acrosomal proteins (the “acrosome reaction”) before they can penetrate the zona pellucida [1,2,3,4,5,6], and mouse spermatozoa traversing the cumulus oophorus begin the acrosome reaction before they reach the zona pellucida [34]. Consequently, the cumulus mass may rescue the sterile phenotype from ablation of genes such as of *Pmis2* by promoting the responsiveness of cells that otherwise are refractory to the spontaneous acrosome reaction needed to fertilize cumulus-free eggs.

How might SMA20/PMIS2 facilitate sperm transport in the female tract and potentiate the acrosome reaction? The sole prior study of PMIS2 [31] reported its initial detection in mouse spermatozoa (by two-dimensional IEF/SDS-PAGE), cDNA cloning, and gene knockout phenotype, but it did not localize the protein in spermatozoa or characterize its physicochemical or immunological properties. The mouse cDNA encodes a 96-residue polypeptide lacking an AP-rich N-terminal sequence and constitutes the only empirical evidence not only for existence of a *Pmis2* structural gene in mouse but also for the identification of orthologs in the genomes of other species. Our independent discovery of SMA20 as a 126-residue polypeptide in pig spermatozoa and the characterization of its biochemical and immunochemical properties, species diversity, and localization provide new information relevant to the protein’s essential function in fertilization. Specifically, we can now infer the nature of the protein’s association with and topology within subcellular membranes of spermatozoa and, in turn, the ways it might interact with other proteins during sperm development and fertilization.

First, SMA20 is an integral membrane protein, based on its isolation from highly purified, salt-washed membrane preparations and quantitative extraction from membranes with non-ionic detergent, as well as the presence downstream in its polypeptide sequence of two relatively conserved hydrophobic segments that likely serve as membrane anchors. Second, SMA20/PMIS2 is most likely not a Type I (N-terminus out) single-pass membrane protein, based on the presence of the N-terminal AP-rich domain rather than a cleavable N-terminal signal peptide that would normally target a Type I protein to the endoplasmic reticulum (ER) by the SRP-dependent pathway [35,36]. Instead, downstream hydrophobic segments of polypeptides lacking a cleavable signal sequence can direct targeting to the ER membrane by the SRP-independent pathway [36,37] and then remain as non-cleaved signal–anchor sequences in the mature protein [38,39,40], with an extraordinary variety of possible resultant membrane topologies [41,42,43]. Figure 10 illustrates three candidate topologies consistent with our immunolocalization results.

Either of the two SMA20 hydrophobic segments (TMS1 or TMS2) could serve as a signal–anchor sequence responsible for directing the protein to the ER and produce each of the three candidate topologies. However, our immunolocalization results collectively favor topologies II and III. We readily detected specific SMA20 immunoreactivity with both anti-P-rich and anti-loop antibodies by immunofluorescence on fixed/permeabilized spermatozoa, but not on motile, live-labeled cells, indicating that neither epitope is surface-exposed. Thus, TopI is unlikely because it orients the loop region at the extracellular surface of the plasma membrane, where it would presumably be accessible to an antibody on live-labeled cells. Also, immuno-EM localized the AP-rich domain to the inner leaflet of the plasma membrane and outer acrosomal membrane, providing further evidence for a cytoplasmic orientation of the N-terminus. Consistent with that result, the “positive-inside rule”, which states that the cytoplasmic sequence generally carries a more positive charge than exoplasmic sequence [43,44], also favors a N_cyto_/C_exo_ orientation for SMA20; however, the location of TMS2 very near the C-terminus could confound this interpretation [41]. Finally, the likelihood that both TMS1 and TMS2 embed in the lipid bilayer favors topology III, with well-established precedent for the proposed “dipping” of TMS1 into the bilayer on the cytoplasmic side [43,45,46]. Regardless, both topology II and topology III position the AP-rich and loop regions for interaction with other proteins in the narrow cytoplasmic space between the plasma and outer acrosomal membranes.

The profound species variation in the sizes and amino acid sequences of AP-rich and loop regions suggests that SMA20/PMIS2-interacting proteins, whatever they may be, likely also differ between species. Short sequence motifs in the AP-rich region, though different or even absent in some species, provide an opportunity for protein–protein interactions. For example, in many species, canonical Src homology 3 (SH3) consensus motifs, PXXP, including PQTP in the mouse, could serve as anchoring sites for cell signaling components. Proline-rich ligands offer great versatility in signaling pathways because their interaction is largely hydrophobic, and SH3 domains can bind polyproline sequence in both directions [47]. Various studies support the general idea that proline-rich proteins may function as an “adaptor” system that assembles multiple proteins into larger complexes [48]. Indeed, species differences in the AP-rich region suggest that the mere presence of SH3-binding or other proline-rich motifs, rather than overall amino acid sequence identity, may be more important for SMA20 function. Furthermore, in contrast to the potentially slow binding kinetics of highly specific protein–protein interactions, proline-rich motifs can bind rapidly and strongly but promiscuously to a large range of proteins by means of an extended “sticky arm” frequently found at amino- or carboxy-termini [49]. The presence of alanine modulates the stiffness of proline-rich regions, thus conferring more flexibility to the segment, and phosphorylation may also regulate binding activity [48,49,50]. Finally, proline-rich regions participate in synaptic vesicle endocytosis [51], which implicates proline motifs in protein trafficking and the fusion events of closely apposed membranes, such as occurs in exocytosis of the acrosome.

Collectively, our findings suggest that SMA20/PMIS functions as an adapter for the assembly of protein complexes, components of which could include gene products already implicated in the normal trafficking and localization of ADAM3. The membrane association of SMA20/PMIS2 hydrophobic segments and cytoplasmic orientation of AP-rich and loop regions position them for interaction with other components, both within and at the cytoplasmic surfaces, respectively, of the sperm outer and peri-acrosomal membranes. Defects in the formation of membrane protein complexes caused by *Pmis2* loss-of-function may then manifest as the dysregulated capture and release of spermatozoa by the utero-tubal junction [52] and as diminished responsiveness of fertilizing spermatozoa to spontaneous acrosome reaction in the vicinity of eggs lacking a cumulus oophorus [31,34].

Species variation in fertilization events necessarily require corresponding species differences in the nature or timing of molecular processes that mediate them. Rapid molecular evolution is a common feature of reproductive proteins in general [53] and of fertilization proteins in particular [29,52], with the adaptive divergence of any one protein potentially driving co-evolutionary divergence of its interacting partners [54]. Known sources of species variation in gene products include protein domain and whole-gene duplications [3,4,5,6,7,8,9,10,11,12,13,14,15,16,17,18,19,53,54,55], rapid sequence divergence by positive selection [29,53,54], concerted evolution by gene conversion [53,54,55], and individual variation in pre-mRNA splicing [56]. Our genome ontogeny results showing that *Sma20*/*Pmis2* arose de novo in the lineage leading to extant placental mammals add this relatively rare mode of new gene formation [57,58] to the repertoire of genetic processes that drive the species diversification of fertilization proteins and cellular events.

## 4. Materials and Methods

### 4.1. Materials

Sources of materials were chemicals, Fisher Scientific (Fair Lawn, NJ, USA), or Sigma Chemical Co. (St-Louis, MO, USA); reagents and protein standards for SDS-PAGE, Bio-Rad Laboratories (Hercules, CA, USA); mass spectrometric grade chemicals and IPG strips for two-dimensional gel electrophoresis, GE Healthcare Bio-Sciences AB (Uppsala, Sweden); EDTA-free protease inhibitor tablets, Roche/Boeringher-Mannheim Biochemicals (Mannheim, Germany); nitrocellulose membrane, GE Healthcare (Piscataway, NJ, USA); PVDF Immobilon P membrane, Millipore Corp. (Billerica, MA, USA); horseradish peroxidase (HRP)-conjugated antibodies, Biosource International (Camarillo, CA, USA), Super Signal West Pico Chemiluminescent Substrate, Pierce Chemical Co. (Rockford, IL, USA); X-ray film, Phoenix Research Products (VWR, Batavia, IL, USA); electron microscopy reagents, Electron Microscopy Sciences (Hatfield, PA, USA). 

### 4.2. cDNA Cloning

We first amplified a partial 3′-end SMA20 cDNA via the 3′-rapid amplification of cDNA ends (3′-RACE). RNA for first-strand synthesis was pig testis total RNA isolated using either the guanidinium-isothiocyanate/acidic-phenol/chloroform extraction method [59] or TRIzol reagent (Invitrogen Corp., Carlsbad, CA, USA). We synthesized the first strand by reverse transcription with SuperScript II reverse (200 units; Invitrogen) primed with an oligo-dT adaptor primer (5′-GACTCGAGTCGACATCGA(T)_17_-3′; RT_3′RACE). Nested PCR using degenerate primers encoding the peptide WEEAYiN (where i may be I or L) consisted of a first-round amplification with forward primer 5′-AACWSCAAGTGGGAGGAGG-3′ (Sense A) and reverse (adaptor specific) primer 5′-GACTCGAGTCGACATCG-3′ (PCR_3′RACE), followed by re-amplification, with nested forward primer 5′-AAGTGGGAGGAGGSCTACMTSAAC-3′ (Sense B) and the reverse primer PCR_3′RACE, using the product from the first round diluted 100-fold in water as template. Cycle parameters were initial soak at 94 °C for 3 min to activate “hot start” DNA polymerase (TAQ Platinum; Invitrogen Corp., Carlsbad, CA, USA); 35 cycles of denature at 94 °C for 30 s, anneal 50–58 °C 30 s (optimized in 2 °C increments depending on primers used), and extend 72 °C 150 s (1st round) or 60 s (2nd round, for products < 1 kb); and a final soak at 72 °C for 5 min to add 3′-A’s for T-A cloning. SMA20 PCR products were T-A cloned into pGEM-T Easy vector (Promega Biosciences, Madison, WI, USA), and cDNA inserts (111–559 bp) from 8 different clones sequenced in both directions. We then cloned the SMA20 5′-end by 5′-RACE, using cDNA template synthesized by SMART technology (Switching Mechanism At the 5′ end of RNA Template; Clontech Laboratories, Mountain View CA) from 1.5 μg of testis total RNA. Gene-specific, 5′-RACE antisense primers (Appendix A) designed from the partial SMA20 3′-cDNA sequence (Figure 1A) spanned nucleotides 15–45 (primer 1), 104–140 (primer 2), 312–338 (primer 3), 35–56 (primer 4), 38–57 (primer 5), and 61–82 (primer 6). PCR with the forward SMART UPM adaptor sense primer and SMA20-specific antisense primers consisted of 25 amplification cycles with the same initial and final soaks and denature step parameters as for 3′-RACE, but with annealing at 60, 65, or 68 °C for 30s depending on the antisense primer used, and extension at 72 °C for 180 s per cycle. SMA20 cDNA inserts (280–531 bp) from 5 different clones were isolated and sequenced in both directions. Finally, to determine a definitive, full-length SMA20 sequence, we repeated 3′- and 5′-RACE amplification, using gene-specific primers and mix of Taq and PFU polymerases (8:1 ratio, [25]), to minimize DNA synthesis errors. Sequences of the resultant 280–531 bp RACE products spanned nucleotides 1–487 (5′-end) and 303–786 (3′-end) of the final, composite SMA20 cDNA.

### 4.3. Northern Blotting

We resolved TRIzol-isolated total RNAs (5 μg each) on 1% formaldehyde–agarose gels; turbo-blotted and UV-crosslinked to MagnaCharge nylon membrane (Micron Separation Inc., Westborough, MA, USA); and hybridized according to a modified Church protocol [60] consisting of pre-hybridization 65 °C 1 h, hybridization with α-^32^P-labeled probe 65 °C 16 h, stringency washes with 1X SSC (150 mM NaCl, 15 mM Na-citrate, pH 7.0) containing 0.1% SDS at 23 °C 2 × 15 min, 65 °C 2 × 30 min and 1 × 20 min, and film exposure at −70 °C with two intensifier screens.

For detection of SMA20 transcript in pig tissues, we prepared an SMA20 probe by random primer-labeling (Ready-to-go DNA labeling beads, GE Healthcare, Chicago, IL, USA) 50 ng of 395 bp SMA20 5′-end cDNA fragment with 50 μCi of α-^32^P-dCTP and removing unincorporated nucleotides on Sephadex G-50 spin columns (Roche Quick Spin, Millipore Sigma, St Louis, MO, USA). To determine if the 1.1 kb SMA20 transcript spanned the full length of the composite cDNA assembled by 3′- and 5′-RACE, we hybridized blots of testis RNA with 5′- and 3′-end oligonucleotide probes (5′-AGAGCCAAGCCCACAGAGAGCCTC-CCCTTGT-3′, and 5′-AGCCAACCAGTTCGGCCTGAGTTGAGGTAGG-3′; 100 ng each) prepared by 5′-end-labeling with 150 μCi of γ-^32^P-dCTP, using 10 U of T4 polynucleotide kinase (Promega) for 10 min at 37 °C. The reaction was terminated via the addition of EDTA to 10 mM and heat-inactivating the enzyme at 78 °C for 1 min, and unincorporated nucleotides were removed as for random primer-labeled probes. SMA20-probed blots were stripped and re-probed for S16 ribosomal RNA to verify the equal loading of RNAs.

### 4.4. Database Queries

To identify putative SMA20 orthologs, we conducted BLASTp query (5 November 2024) of NCBI nr protein and nucleotide databases, supplemented with gene name and homology-based searches to identify the genomic loci (NCBI Gene database) of porcine SMA20 and its putative orthologs in species lacking annotated *Pmis2*. We also queried the NCBI Gene database (10 November 2024) to retrieve the pig and other species’ syntenic regions spanning *Atp4a*–*Haus5* and conducted pairwise dotplot comparisons (MegAlign program of the Lasergene 15 software suite; DNAStar, Madison, WI, USA) to identify SMA20/PMIS2 loci and to align deduced amino acid sequences of predicted ORFs.

### 4.5. Sperm Preparation and Membrane Isolation

To prepare the spermatozoa for microscopy and subcellular fractionation, we washed cells in extended semen of fertile boars (obtained from the Texas Tech Swine Center, New Deal, TX, USA) via dilution in phosphate-buffered saline (PBS; 20 mM NaPO4 pH 7.4, 150 mM NaCl) and centrifugation (400× *g*, 8 min) to remove the extender. We then resuspended the loose pellet in PBS at 25–30 × 10^6^ cells/mL [17]). We then isolated a particulate fraction enriched in sperm plasma membranes (= triple-washed membranes, TWM) and prepared detergent-resistant membranes (=lipid rafts) also as previously described [22]. Briefly, we selectively disrupted the plasma membranes of washed spermatozoa via N_2_ cavitation at 650 psi; centrifuged at 1000× *g* to remove sperm heads and tails and centrifuged at 10,000× *g* to remove subcellular organelles and fragments; and then recovered a membrane-containing particulate fraction via re-centrifugation at 100,000× *g*. The membrane pellet was then washed thrice via consecutive resuspension in buffer containing 1 M NaCl and centrifugation at 100,000× *g* to remove electrostatically associated peripheral proteins and thereby produce a particulate fraction enriched with integral membrane proteins.

### 4.6. Antibody Production

For the immunochemical characterization of SMA20, we produced two monospecific peptide antibodies to the predicted proline-rich (“P-rich”) and loop (“loop”) regions of SMA20 (Pacific Immunology, Ramona, CA, USA). The P-rich antigen comprised amino acids 21–52 (32 residues), and the loop antigen comprised amino acids 82–96 (15 residues), with the latter being located between the two putative transmembrane domains (Figure 1). Each antibody was then antigen affinity-purified from the pooled sera of two rabbits on columns coupled with the respective peptides for the isolation of domain-specific antibodies to zonadhesin [15,61].

### 4.7. Electrophoresis and Western Blotting

For Western blotting, we resolved proteins on straight 15% or 8–15% linear gradient gels (one-dimensional SDS-PAGE) or sequentially on pH 3–10 strips and straight 12% gels (two-dimensional IEF/SDS-PAGE) [22]. On one-dimensional gels, we loaded 20 μg of protein per lane of triple-washed membranes (TWM), or proteins from 5 × 10^6^ spermatozoa per lane of detergent fractions obtained on sequential extraction with 1% Triton X-100 and 1% SDS [21]. On two-dimensional electrophoresis, we applied 50 μg of solubilized TWM protein per IEF strip. For immunodetection, blots were incubated overnight at 23 °C with anti-loop or anti-P-rich antisera (1/5000) or with affinity-purified antibodies (anti-loop, 0.4 µg/mL; anti-P-rich, 0.02 µg/mL). To assess antibody binding specificity, we blocked 10 µL of affinity-purified antibody by incubating 1 h at 23 °C with an excess of the cognate peptide (11 µg of loop in 11 µL PBS or 8 µg of P-rich in 8 µL PBS), or similarly blocked 5 µL of serum (loop or P-rich) with 5 µg of cognate peptide.

### 4.8. Immunofluorescence

All steps for immunofluorescence were carried out at 23 °C, as previously described [61]. For light microscopy of fixed, permeabilized cells, we gently smeared 20 μL of washed sperm suspension (20–30 × 10^6^ cells/mL in PBS) onto cleaned glass slides, air-dried, and then fixed in methanol for 30 min. We then localized SMA20 in spermatozoa by immunofluorescence using anti-loop (1.1 μg/mL) or anti-P-rich (0.8 µg/mL) affinity-purified antibodies diluted in 10% (*v*/*v*) heat-inactivated goat serum (HIGS)/PBS (30 min incubation) and assessed antigen-binding specificity by comparing to antibodies preabsorbed with their cognate peptides or normal rabbit IgGs (1.1 or 0.8 µg/mL in HIGS/PBS). After washing the slides with PBS, bound antibody was detected with Alexa 594-conjugated goat anti-rabbit IgG (0.5 μg/mL in 10% (*v*/*v*) heat-inactivated pig serum/PBS; 30 min incubation). Plasma membranes or acrosomes were then lectin-labeled for 30 min with FITC-conjugated wheat germ agglutinin (WGA; 10 μg/mL) or peanut agglutinin (PNA; 2 µg/mL), respectively. After a final wash in PBS, slides were mounted with coverslip and Fluoromount G and viewed by epifluorescence and phase-contrast microscopy (400× magnification).

To examine whether SMA20 is located on the plasma membrane, we performed immunofluorescence on living, motile spermatozoa by rocking cells in suspension with the loop (1.1 μg/mL) or P-rich (0.8 μg/mL) AP antibodies or normal rabbit IgGs (NR IgGs, 1.1 or 0.8 µg/mL) in PBS containing 10% (*v*/*v*) HIGS (30 min at 23 °C). Non-bound antibody was removed by centrifugation (200× *g*, 1 min), and spermatozoa were gently smeared on slides and air-dried. Bound antibody, plasma membranes, and acrosomes were detected as for IF on fixed cells. Alternatively, incubation with the secondary antibody was also carried out in tube as above, and then spermatozoa were smeared on slides and air-dried. Both motility (light microscopy) and viability (Live/Dead assay, Invitrogen) were assessed before and after the incubation with primary antibodies. Although some sperm agglutination was observed after 30 min of incubation, spermatozoa were motile and viable (80% viable cells in average; n = 2).

### 4.9. Immunoelectron Microscopy

We characterized ultrastructure of SMA20 localization in spermatozoa by immunoelectron microscopy, using affinity-purified (AP) loop and P-rich antibodies according to both post- and pre-embedding immunolabeling protocols [62,63]. For post-embedding immunolabeling, washed spermatozoa were fixed on ice with 4% paraformaldehyde and 0.25% glutaraldehyde in 0.1 M Sorenson phosphate buffer pH 7.2 (Karnovsky’s fixative); dehydrated through an ethanol series; post-fixed (or not) with 1% osmic acid to stabilize lipids and, hence, preserve membranes; and embedded in LR White resin. The polymerized blocks were trimmed with razor blades and sectioned to 88 nm on an RMC PowerTome ultramicrotome (Boeckler Instruments, Tucson, AZ, USA). Thin sections were mounted on nickel grids ± support film and immunostained with AP anti-loop (11 µg/mL), anti-P-rich (8 µg/mL), normal rabbit IgGs (NR IgGs, 11 µg/mL and 8 µg/mL, respectively), or preabsorbed AP anti-loop and anti-P-rich, as previously described for Western blot and immunofluorescence experiments. For most of the post-embedding immunolabeling experiments, we “unmasked” antigens with an additional “Heat-Induced Antigen Recovery” step (Zymed Laboratories Inc., South San Francisco, CA, USA) prior to immunolabeling to facilitate the binding of the antibody. To block irrelevant binding sites on both spermatozoa and resin, the grids were incubated with Aurion goat blocking solution containing 0.1% Tween 20. After immunostaining, the sections were washed in PBS, fixed in 1% glutaraldehyde in 0.1 M sodium phosphate buffer, rinsed with water, and stained with uranyl acetate and Reynolds lead citrate.

For pre-embedding immunolabeling, washed, intact spermatozoa were fixed on ice for 20 min in two volumes of 2% glutaraldehyde in 0.1 M Sorenson phosphate buffer pH 7.2. To block free aldehydes, suspension of fixed spermatozoa was incubated with PBS containing 50 mM glycine for 10 min, centrifuged at 200× *g* for 2–3 min, and resuspended in PBS. Immunolabeling of cell suspension was carried out as for the immunofluorescence of live labeled spermatozoa, but using antibodies and buffers/solutions as detailed for post-embedding immunolabeling procedure. Alternatively, sperm suspension was diluted 5-fold with 0.25% Triton X-100 and extracted on ice for 20 min prior to fixation. After the final wash in PBS, the sperm pellet was fixed in 1% glutaraldehyde in 0.1 M Sorenson phosphate buffer pH 7.2, post-fixed with 1% osmic acid, dehydrated through an ethanol series, and embedded in EPON resin (LX 112). Thin sections were mounted on copper grids ± support film and stained with uranyl acetate and Reynolds lead citrate.

We viewed immunolabeled spermatozoa on a Hitachi H-7650 transmission electron microscope (Hitachi High Technologies America, Inc., Pleasanton, CA, USA) and quantified specific immunolabeling by counting the gold particles associated with the peri-acrosomal plasma membrane (n = 40 fields) or sperm head (n = 20 fields), as well as in the entire corresponding field. Then, we compared the ratios to those calculated for negative controls. Data from the immunoelectron microscopy studies (number of gold particles in sperm apical head/total in the field) were expressed as mean ± SEM. Comparisons between group means of immunolabeled spermatozoa and controls with preabsorbed antibodies were performed with Microsoft Excel, using the Two-Sample Unpaired *t*-Test and assuming equal variances.

## Figures and Tables

**Figure 1 ijms-25-03652-f001:**
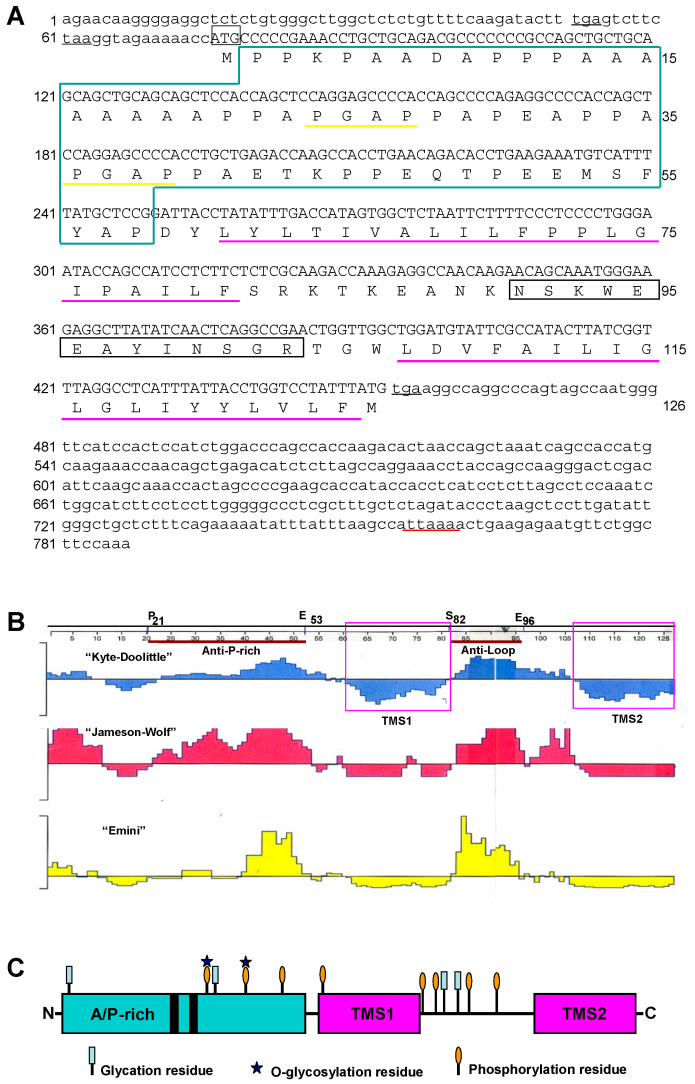
Sequences of SMA20 composite cDNA and its predicted 126 amino acid polypeptide. (**A**) Nucleotide and deduced amino acid sequences of the 786-bp SMA20 composite cDNA. The tryptic peptide obtained by de novo sequencing is boxed in black. Numbers on the left denote positions in the nucleotide sequence, and numbers on the right denote positions in the amino acid sequence. The 5′- and 3′-UTRs (lowercase letters) flank a 378 nt coding sequence (upper case letters) with the predicted initiator methionine (boxed in gray) embedded in a good Kozak consensus (AcAGcccccg; Ref. [23]), a non-canonical ATTAAA polyadenylation signal (red underline) specifying transcript termination, and in-frame stop codons (black underlines) in both UTRs. Features of the encoded protein include, in order, a region of AP-rich sequence (boxed in green) that includes two PGAP putative SH3 domain binding sites (yellow underlines) but no contiguous hydrophobic amino acids expected for an N-terminal signal peptide [24], followed by two predicted transmembrane segments (pink underlines). (**B**) Predicted SMA20 properties. Shown are the hydrophilicity (“Kyte–Doolittle”), antigenicity (“Jameson–Wolf”), and surface probability (“Emini”) inferred from the SMA20 protein sequence. The horizontal ruler denotes the amino acid position, and the vertical axes represent arbitrary units. Peptides used for antibodies (overlined in red) spanned amino acids P^21^-E^53^ (anti-P-rich) and S^82^-E^96^ (anti-loop) between the two putative transmembrane segments (TMS; boxed in pink). (**C**) SMA20 structural modules. Asterisks and small ovals and rectangles mark locations of predicted sites for posttranslational modification as indicated in the alanine- and proline-rich (“A/P-rich”) region containing two predicted SH3 binding motifs (PGAP, black bars), and in the loop region flanked by predicted transmembrane segments (“TMS1/2”).

**Figure 2 ijms-25-03652-f002:**
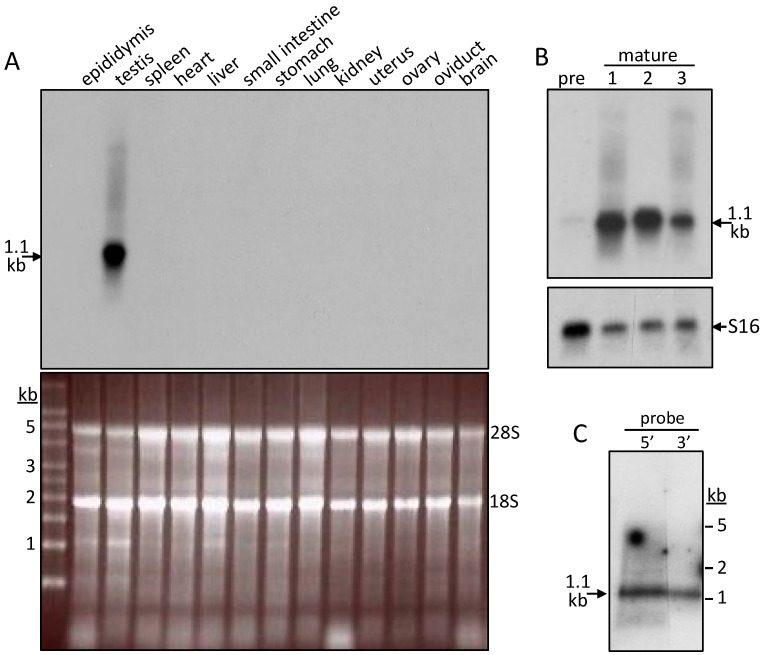
Tissue specificity of SMA20 mRNA expression. (**A**) Upper panel: Detection of SMA20 mRNA on Northern blot of total RNAs (5 μg/lane) isolated from pig tissues via hybridization of a 5′-end cDNA probe (nts 1–395; 2-day exposure). All tissues except epididymis and testis came from a female pig. Lower panel: The ethidium bromide-stained gel before blot transfer, confirming quality and equal loading of the total RNA preparations. (**B**) Northern blot confirmation of the full-length SMA20 composite cDNA via hybridization of oligonucleotides (30 mer each) complementary to 5′- and 3′-end SMA20 cDNA to blots of pig testis total RNA (20 μg/lane; 4-day exposure). (**C**) Upper panel: Northern blot detection of SMA20 mRNA (5 μg total RNA per lane) in pig testis from a pre-pubertal boar (“pre”) and from three mature boars (“1–3”) via hybridization with the same SMA20 cDNA probe as for panel A. Lower panel: hybridization of the stripped blot with a probe for S16 ribosomal RNA to verify equivalent loading.

**Figure 3 ijms-25-03652-f003:**
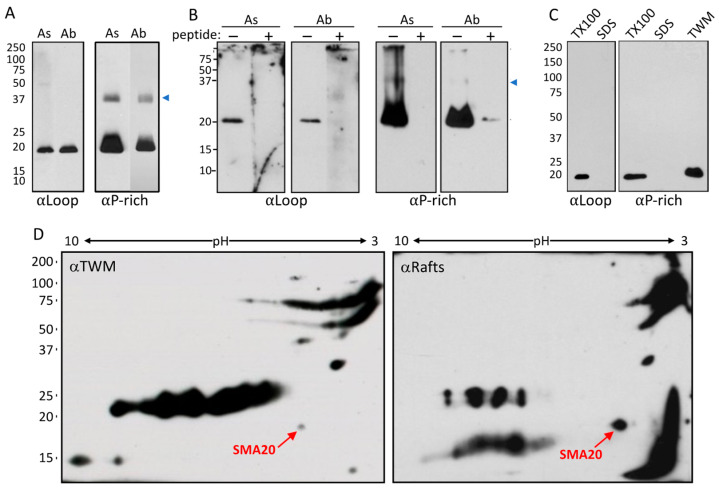
Membrane and lipid raft association of SMA20 in pig spermatozoa. Shown are Western blots probed with raw heteroantisera (“As”) or affinity-purified antibodies (“Ab”) directed against the SMA20 loop or proline-rich regions (“αLoop” or “αP-rich” respectively; panels (**A**–**C**)), or with alloantisera raised against pig sperm TWM or lipid rafts (“αTWM” or “αRafts”; panel (**D**)). (**A**) TWM resolved on 8–15% linear gradient SDS-PAGE (10 μg protein/lane) and detected with αLoop or αP-rich antisera or purified antibodies. The blue arrowhead marks the location of the M*r* 37,000 band recognized by the αP-rich peptide antibody. (**B**) 20 μg of TWM resolved on 15% SDS-PAGE; detected with αLoop or αP-rich antisera or purified antibodies preabsorbed (+), or not (−), with their cognate peptides. Note the robust blocking of immunoreactivity by the peptides. (**C**) SMA20 in proteins extracted sequentially from 5 × 10^6^ spermatozoa with 1% Triton X-100 (TX100) and 1% SDS [26] and then resolved and blotted as for panel (**A**). (**D**) Alloantigens in 50 μg of TWM resolved by two-dimensional gel electrophoresis and detected on blots with αTWM or αRafts alloantisera. Red arrows indicate SMA20 protein (*M*r 20,000 and p*I* 4.3).

**Figure 4 ijms-25-03652-f004:**
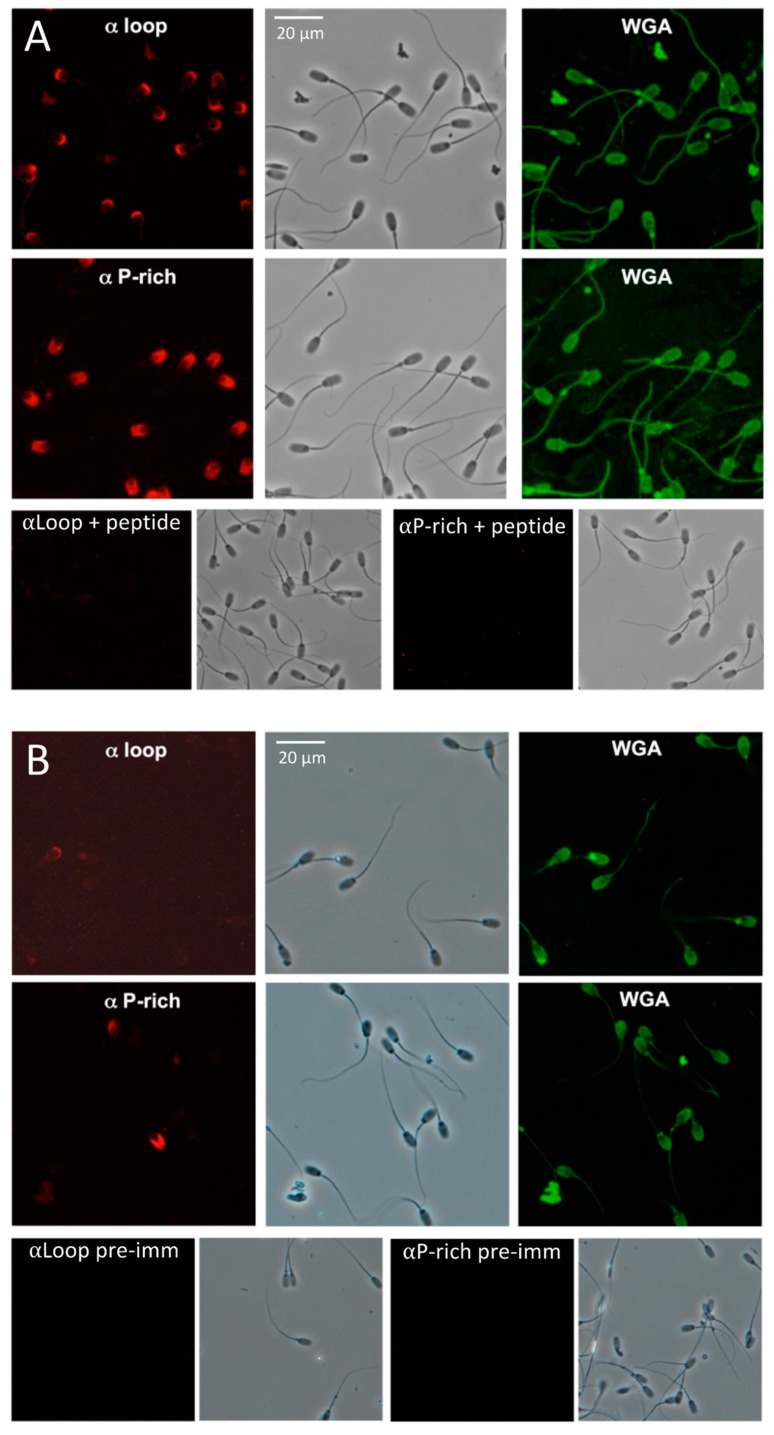
Localization of SSMP20 protein on pig spermatozoa by immunofluorescence. Shown are paired fluorescence and phase contrast images of identical microscopic fields (400× magnification). WGA = labeling with wheat germ agglutinin to detect plasma membrane. (**A**) Detection of SMA20 in MeOH-fixed (=membrane permeabilized) spermatozoa. Red fluorescence depicts SMA20 immunoreactivity detected with affinity-purified antibodies directed against the SMA20 loop (“αLoop”, 1.1 μg/mL) or proline-rich (“αP-rich”, 0.8 μg/mL) regions, respectively, and green fluorescence depicts WGA detection of the acrosome. (**B**) Detection of SMA20 on live-labeled spermatozoa.

**Figure 5 ijms-25-03652-f005:**
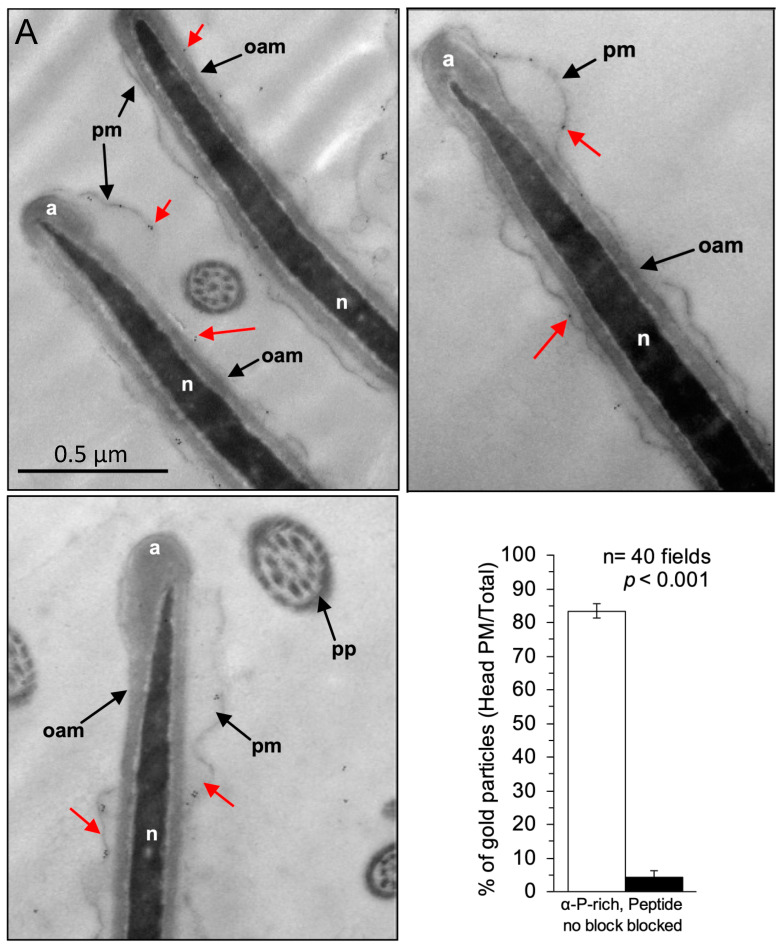
Ultrastructural localization of SMA20 by immuno-EM with αP-rich affinity-purified antibody. Shown are multiple fields of cross-sections through heads of mature spermatozoa labeled after embedding and sectioning. (**A**) Cells post-fixed with osmium tetroxide. Red arrows mark SMA20-associated gold nanoparticles localized in apposition to the inner leaflet of the plasma membrane (pm) overlying the anterior head, with its ruffled appearance characteristic of spermatozoa in TEM [26], as well as much fewer on the outer acrosomal membrane (oam). The bar graph summarizes relative quantification of gold particles’ specific association with the head plasma membrane, evident as nearly complete blocking of immunoreactivity by cognate peptide in 40 randomly chosen microscopic fields. a = acrosome; n = nucleus. (**B**) Cells without osmium post-fixation. The bar graph summarizes relative quantification of specific labeling as for panel (**A**).

**Figure 6 ijms-25-03652-f006:**
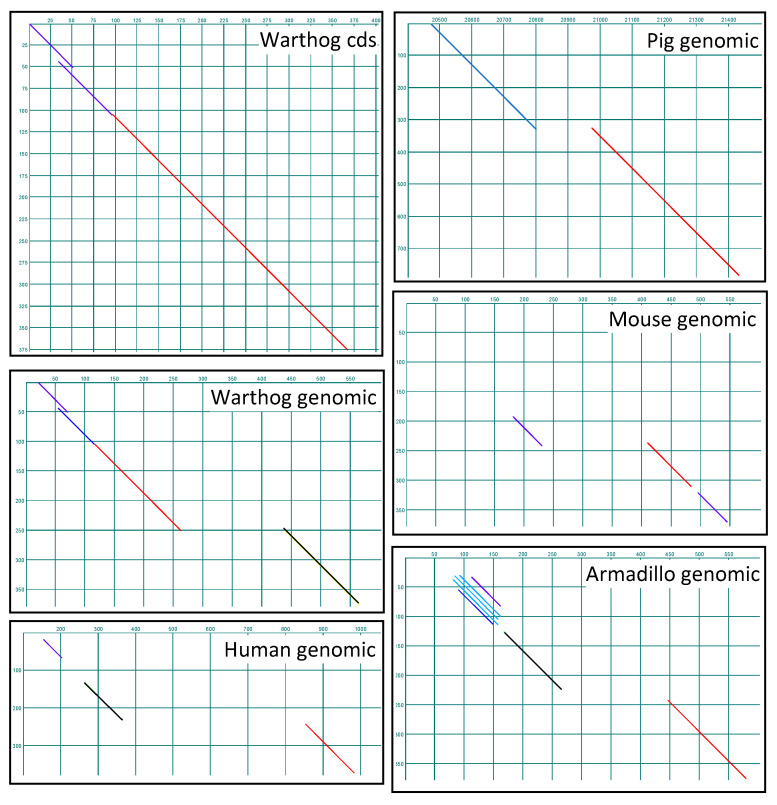
SMA20 is the pig ortholog of PMIS2. Shown are pairwise dot-plot comparisons of the SMA20 cDNA sequence (vertical dimension in all panels) to the warthog coding sequence (cds) and, as indicated, warthog, pig, mouse, human, and armadillo genomic sequences between *Atp4a* and *Haus5* in their respective species (horizontal dimensions).

**Figure 7 ijms-25-03652-f007:**
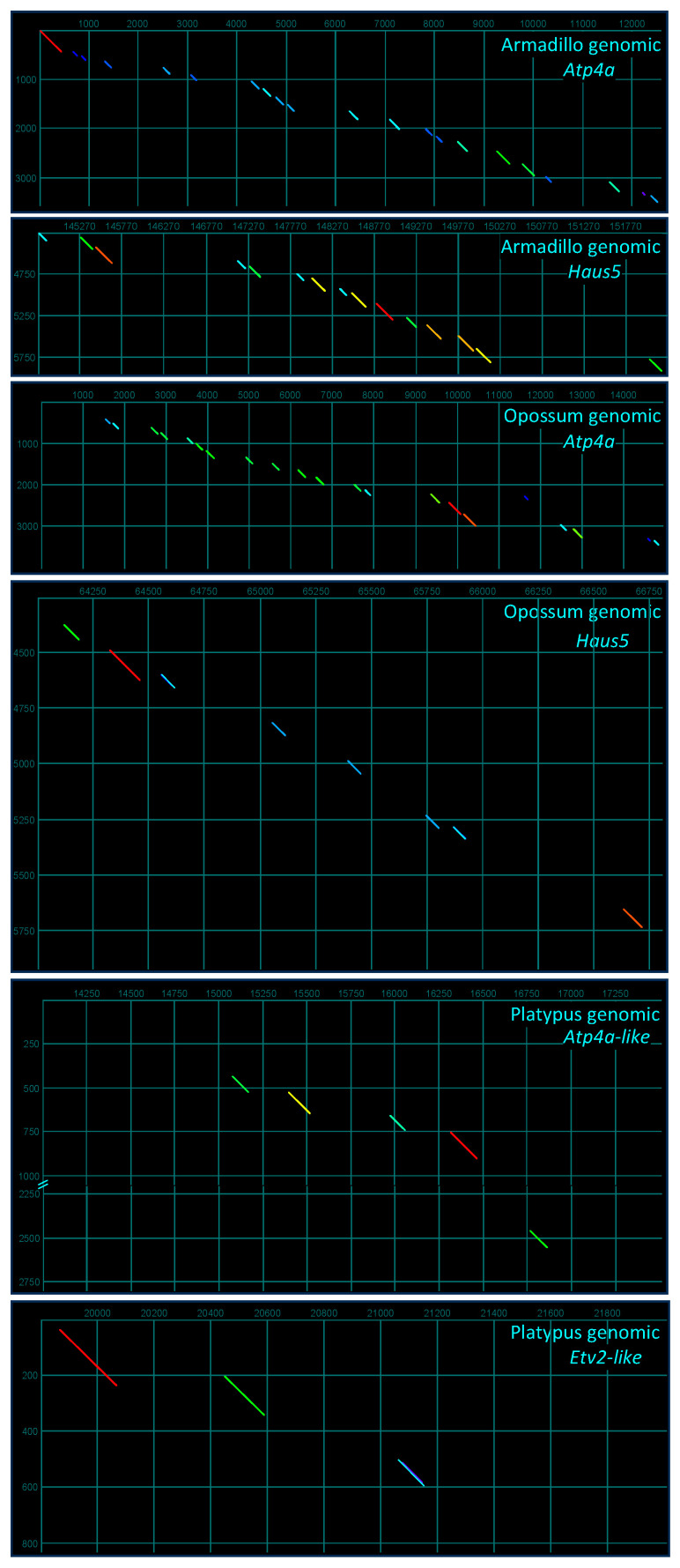
SMA20/PMIS2 is unique to Eutheria. Shown are dotplot comparisons constructed with a concatemeric assembly of armadillo reverse-complemented *Atp4a*, *Pmis2*, and *Haus5* coding sequences (cds) as the vertical dimension for interrogation of armadillo and opossum genomic loci, and a comparable assembly with reverse-complemented *Atp4a*, *Pmis2*, and reverse-complemented *Etv2* cds for interrogation of platypus genomic loci.

**Figure 8 ijms-25-03652-f008:**
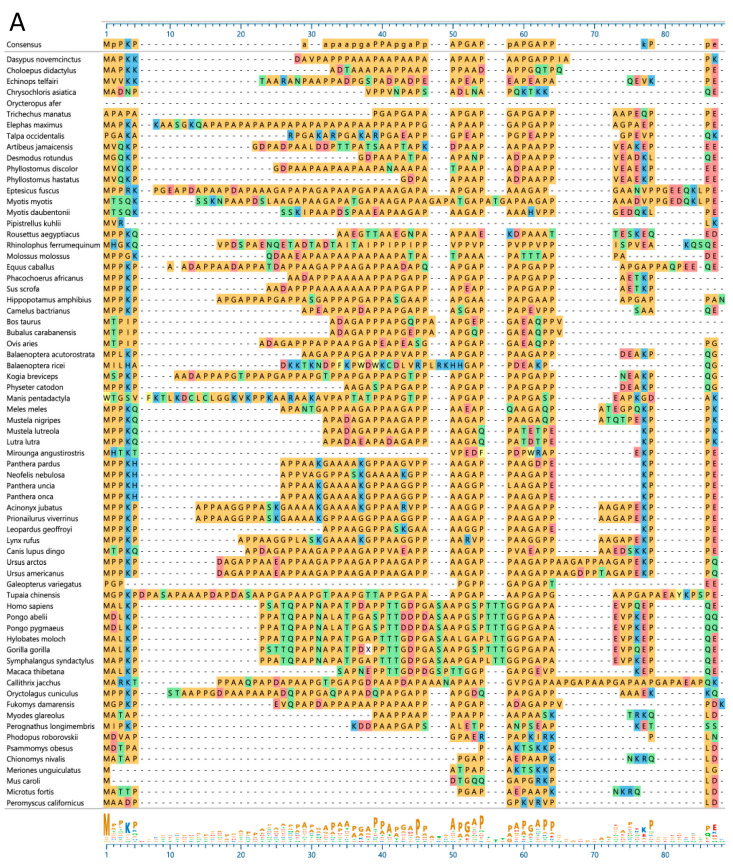
Multiple alignment (MUSCLE) of pig (*Sus scrofa*) SMA20 polypeptide sequence to PMIS2 orthologs from 69 species representing 17 of the 19 extant orders of placental mammals (Afrosoricida, Carnivora, Cetartiodactyla, Chiroptera, Cingulata, Dermoptera, Eulipotyphla, Lagomorpha, Perissodactyla, Pholidota, Pilosa, Primates, Proboscidea, Rodentia, Scandentia, Sirenia, and Tubulindentata; missing are Hyracoidea and Macroscelidea). (**A**) Alignment of N-terminal region sequences. The bottom seven species are myomorph rodents. (**B**) Alignment of TMS1 and TMS2 region sequences showing the sequence variation in segments upstream of the relatively more conserved transmembrane segments (red overlines). Colored shading denotes similar amino acid side chain chemistries: light yellow = aromatic, dark yellow = hydrophobic/aliphatic, green = hydrophilic, blue = basic, red = acidic.

**Figure 9 ijms-25-03652-f009:**
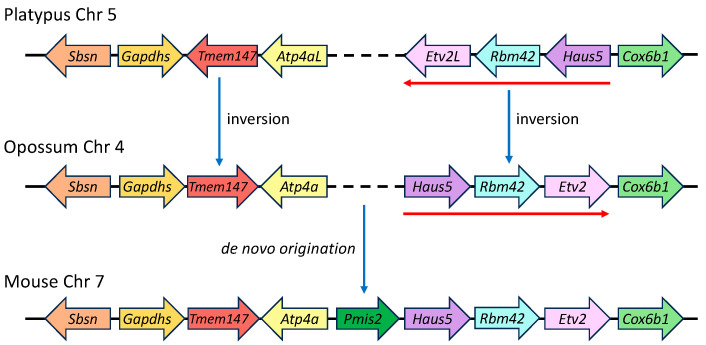
Genomic ontogeny of SMA20/PMIS2. Shown is the comparative synteny of regions encompassing the four upstream and four downstream genes immediately flanking *Pmis2* in the mouse genome. The compared *Sbsn*–*Cox6b1* regions span 140, 146, and 62 kb in the mouse, opossum, and platypus genomes, respectively (loci not drawn to scale). Blue arrows denote major changes that occurred in the evolution of Mammalia, including gene inversions associated with the divergence of Theria and Prototheria (with inverted three-gene cassette denoted by red arrows), as well as de novo genesis of *Pmis2* associated with the divergence of Metatheria and Eutheria.

**Figure 10 ijms-25-03652-f010:**
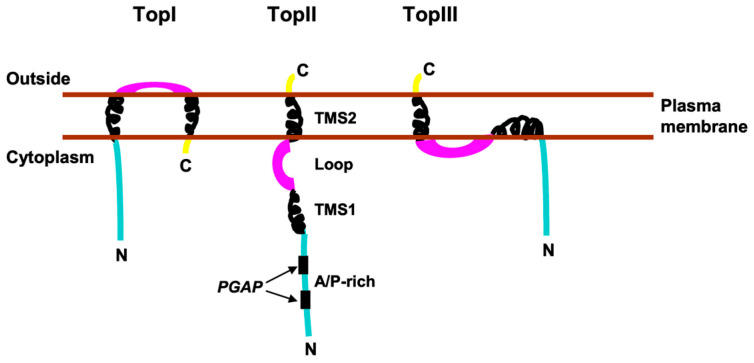
Candidate topologies (TopI-III) of SMA20 in the head of mature, ejaculated boar spermatozoa. All three topologies reflect cytoplasmic orientation of the AP-rich region in accordance with immunolocalization results, with “Outside” being extracellular or acrosomal lumen for SMA20 in the peri-acrosomal plasma membrane or outer acrosomal membrane, respectively. Squiggly lines depict the two presumably alpha helical hydrophobic segments, TMS1 and TMS2, proposed to span or embed in the lipid bilayer, as shown, and the black bars mark locations of possible SH3 binding motifs (*PGAP*) in the AP-rich region.

## Data Availability

The pig SMA20 cDNA sequ4nce will be uploaded to NCBI.

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
