# Peer review of "SMA20/PMIS2 Is a Rapidly Evolving Sperm Membrane Alloantigen with Possible Species-Divergent Function in Fertilization"

_ijms, 2024, doi:10.3390/ijms25073652_

Round 1

Reviewer 1 Report

Comments and Suggestions for Authors

In their manuscript „SMA20/PMIS2 is a rapidly evolving sperm membrane alloantigen with possible species-divergent function in fertilization”, Cormier and colleagues describe the sequence, structure, localization and ontology of the SMA20 pig ortholog of the murine PMIS2. The authors identified the transcript sequence via combined 5’- and 3’-RACE, and described the tissue distribution via Northern Blotting. The localization in the plasma membrane was shown via immunofluorescence and - in more detail - via immunoelectron microscopy. Finally, the authors resolve the ontogeny of SMA20 over different species. The discussion section is a well-written in-depth analysis of the results achieved, and adds further insights in the putative role of SMA20 as an adapter protein for protein complexes involved in the ADAM3 pathway.

Altogether, the manuscript was a well-written, thoughtful study, which was a pleasure to read. The introduction is appropriate and leads towards the aim of the study, the methodology is described in great detail and the results seems sound. The discussion is comprehensive, and the conclusions are covered by the results. I read the manuscript thrice to find any points which could be clarified or improved, but did find none - except for the minor recommendation to maybe introduce subheadings in the Material and Methods section for a better separation of the individual experimental approaches.

Finally, I want to thank the authors for sharing their results with the scientific community. Reading this manuscript was delightful. Best regards.

Author Response

We thank the reviewer for her/his thoughtful comments, and have added subheadings to the Materials and Methods section as recommended.

Reviewer 2 Report

Comments and Suggestions for Authors

The authors documented a gene/protein associated with the sperm membrane and clarified its function. Here are some suggestions to enhance the manuscript.

1.     Please explain how to prepare the TMW.

2.     For the tissue distribution experiment, which sex of animals did you use to obtain the tissue?

3.     For figure 2C, the contrast appears too high to me.

4.     What are the αloop+peptide and αP-rich+peptide in Figure 4? Please describe them in the figure caption.

5.     Please provide scale bars in Figures 4 and 5.

6.     For the detection of SMA20 on live-labeled spermatozoa with immunofluorescence, the authors showed less staining in the live spermatozoa when compared with the fixed sample. How can you be confident that this difference is not due to the experimental design? Do you have any positive control for the live spermatozoa staining?

7.     Line 251, “Gapdhs (gene encoding germ cell-specific GAPDH), please provide a reference for this statement.

8.     Please remove the results explanation from the figure caption. It should be in the results section.

9.     Please rewrite or reorganize the Materials and Methods section. At least provide subtopics.

10.  Did the authors submit the sequence to the database?

Author Response

We appreciate the reviewer's thoughtful comments, and have responded to them as follows.

1. Added text summarizing the method for preparing TWM as requested.

2. Thank you for catching this omission. We added the statement "All tissues except epididymis and testis came from a female pig." to the legend of Figure 2.

3. We rearranged the panels of Figure 2 to improve logical flow and swapped out the flawed images. We don't know how the images came to have such high contrast, and thank the reviewer for noticing.

4. Information added to the Figure 4 legend as requested.

5. Scale bars added to panels 4A, 4B, 5B as requested.

6. The antibodies detected SMA20 on only 5-10% of the WGA positive, live-labeled cells (Results section 2.4), but fluorescence was strong on the cells that did label (Figure 4B). Thus the labeling method itself worked well, but the SMA20 epitopes were inaccessible on most cells. In effect, the labeling of those relatively few, damaged cells served as a positive control for the experiment.

7. Reference added as requested.

8. We deleted results explanation from the figure legends and moved non-repetitive language to the Results section as requested. 

9. Re-written Materials and Methods includes added subheadings and streamlined text as requested.

10. We will submit the cDNA sequence to Genbank when the manuscript is accepted and add the accession number to the final proof.

Round 2

Reviewer 2 Report

Comments and Suggestions for Authors

The authors answered and improved all my questions. I have no further comments.